# The Associations of Breastfeeding Status at 6 Months with Anthropometry, Body Composition, and Cardiometabolic Markers at 5 Years in the Ethiopian Infant Anthropometry and Body Composition Birth Cohort

**DOI:** 10.3390/nu15214595

**Published:** 2023-10-29

**Authors:** Mathilde S. Heltbech, Cecilie L. Jensen, Tsinuel Girma, Mubarek Abera, Bitiya Admassu, Pernille Kæstel, Jonathan C. K. Wells, Kim F. Michaelsen, Henrik Friis, Gregers S. Andersen, Rasmus Wibæk

**Affiliations:** 1Department of Nutrition, Exercise and Sports, University of Copenhagen, 1958 Copenhagen, Denmarkpernille.kaestel@gmail.com (P.K.); kfm@nexs.ku.dk (K.F.M.); hfr@nexs.ku.dk (H.F.); 2Department of Pediatrics and Child Health, Jimma University, Jimma P.O. Box 378, Ethiopia; tsinuel@yahoo.com; 3Department of Psychiatry, Faculty of Medical Sciences, Jimma University, Jimma P.O. Box 378, Ethiopia; abmubarek@gmail.com; 4Jimma University Clinical and Nutrition Research Partnership (JUCAN), Jimma University, Jimma P.O. Box 378, Ethiopia; bitiyaa@yahoo.com; 5Childhood Nutrition Research Centre, UCL Great Ormond Street Institute of Child Health, London WC1N 1EH, UK; jonathan.wells@ucl.ac.uk; 6Clinical Research, Copenhagen University Hospital–Steno Diabetes Center Copenhagen, 2730 Herlev, Denmark

**Keywords:** breastfeeding, infant, child, anthropometry, body composition, cardiometabolic markers, sub-Saharan Africa, noncommunicable diseases

## Abstract

(1) Background: Breastfeeding (BF) has been shown to lower the risk of overweight and cardiometabolic disease later in life. However, evidence from low-income settings remains sparse. We examined the associations of BF status at 6 months with anthropometry, body composition (BC), and cardiometabolic markers at 5 years in Ethiopian children. (2) Methods: Mother–child pairs from the iABC birth cohort were categorised into four BF groups at 6 months: 1. “Exclusive”, 2. “Almost exclusive”, 3. “Predominantly” and 4. “Partial or none”. The associations of BF status with anthropometry, BC, and cardiometabolic markers at 5 years were examined using multiple linear regression analyses in three adjustment models. (3) Results: A total of 306 mother–child pairs were included. Compared with “Exclusive”, the nonexclusive BF practices were associated with a lower BMI, blood pressure, and HDL-cholesterol at 5 years. Compared with “Exclusive”, “Predominantly” and “Almost exclusive” had shorter stature of −1.7 cm (−3.3, −0.2) and −1.2 cm (−2.9, 0.5) and a lower fat-free mass index of −0.36 kg/m^2^ (−0.71, −0.005) and −0.38 kg/m^2^ (−0.76, 0.007), respectively, but a similar fat mass index. Compared with “Exclusive”, “Predominantly” had higher insulin of 53% (2.01, 130.49), “Almost exclusive” had lower total and LDL-cholesterol, and “Partial or none” had a lower fat mass index. (5) Conclusions: Our data suggest that children exclusively breastfed at 6 months of age are overall larger at 5 years, with greater stature, higher fat-free mass but similar fat mass, higher HDL-cholesterol and blood pressure, and lower insulin concentrations compared with predominantly breastfed children. Long-term studies of the associations between BF and metabolic health are needed to inform policies.

## 1. Introduction

Noncommunicable diseases (NCDs), such as cardiovascular disease (CVD) and type 2 diabetes (T2D), are leading causes of premature death worldwide [1]. Nutrition transitions in low- and middle-income countries (LMICs) have resulted in an increasing occurrence of NCDs, which together with a persistently high prevalence of undernutrition is referred to as the double burden of malnutrition [2,3]. The risk of all forms of malnutrition is reduced by a healthy diet, particularly in the first 1000 days, which promotes healthy growth, development, and immunity, and prevents obesity and NCDs [4,5].

Breastfeeding (BF) is associated with multiple benefits to children [6,7] and has, due to nutritional programming effects, been proposed as a preventive strategy against the double burden of malnutrition and cardiometabolic conditions, including T2D and CVD [2,7,8,9]. The programming mechanisms are still unclear; however, several plausible mechanisms have been proposed, among others the following, which are based on a comparison of breastfeeding with formula feeding [10,11,12,13,14,15,16]. A high weight gain during early life is highly associated with later obesity, and the lower weight gain during the first six months of life observed among breastfed infants may play a role in the lower risk of overweight in later life [10,11]. Breastfed infants also tend to have lower insulin and IGF-1 levels, which may be due to a lower intake of energy and protein [11]. Insulin stimulates fat deposition and development of adipocytes, whereas IGF-1 is important for cell growth and mitogenesis. High levels of these hormones may contribute to obesity by up-regulating adipogenesis and adipocyte differentiation [11,12,13]. A reversed pattern has been observed for cholesterol, leading to the hypothesis that early exposure to the high cholesterol content of breast milk could affect cholesterol metabolism in the long term by down-regulating the cholesterol synthesis [14]. Moreover, breastfeeding seems to be associated with a lowering effect on blood pressure (BP), and breastfed children tend to develop a better appetite regulation [15,16]. Despite the multitude of benefits and recommendation of exclusive breastfeeding (EBF) for the first six months of life, suboptimal BF practices are prevalent and widespread [17]. Approximately 50% of Ethiopian mothers breastfeed exclusively at 6 months, with an even lower prevalence of 34% among sub-Saharan African mothers [18,19].

Body composition (BC) and other cardiometabolic markers are important already in childhood, as they may predict the future risk of overweight, T2D, and CVD [11,20]. Most studies investigating the associations of BF with subsequent overweight, BC, and cardiometabolic markers have been conducted in high- and middle-income countries [2,7,21]. Among the few studies conducted in low-income settings, the results are inconsistent [22]. Furthermore, substantial variations in the definition of BF practices are observed [21]. We aimed to examine the associations of BF status at 6 months with anthropometry, BC, and cardiometabolic markers at 5 years of age in a sub-Saharan setting.

## 2. Subjects and Methods

### 2.1. Study Setting and Participants

Between December 2008 and October 2012, the infant Anthropometry and Body Composition (iABC) birth cohort enrolled mothers residing in Jimma town, Ethiopia, and their apparently healthy newborns, born at term (≥37 weeks of pregnancy) with a birth weight ≥ 1500 g [23,24]. The newborns were examined within 48 h after delivery at Jimma University Hospital and were followed up continuously until 5 years of age, with a total of 11 follow-up visits. In the Oromia region, where Jimma town is located, less than 20% of women are giving birth at a health facility, whereas the rest are delivering their babies at home [25,26]. Despite having a fast-growing economy, Ethiopia is still among the poorest countries in the world [27,28]. However, a big socioeconomic contrast exists between urban and rural areas of Ethiopia [29]. More details about the cohort have been published previously [23,24].

### 2.2. Data Collection

#### 2.2.1. Exposure: Breastfeeding Questionnaires at 6 Months

BF status at 6 months was assessed via questionnaires at the 6-month visit. Infants were assigned to one of four BF status groups: (1) “Exclusive”, (2) “Almost exclusive”, (3) “Predominantly”, and (4) “Partial or none” following previously suggested classifications [30]. Details about the questionnaires and construction of BF groups are described in Text A1.

#### 2.2.2. Outcomes: Anthropometry, Body Composition, and Cardiometabolic Markers at 5 Years

Data collection at 5 years has been described elsewhere [24]. Briefly, height was measured standing to the nearest 0.1 cm using SECA 213 portable height measurer (SECA, Hamburg, Germany). Waist circumference was measured midway between last rib and the iliac crest to the nearest 0.1 cm using nonstretchable measuring tape. Both measurements were measured in duplicates and averaged. Weight was measured to the nearest 1 g by the built-in scale of a BOD POD (Cosmed, Rome, Italy) designed for children and adults. The BOD POD, an air displacement plethysmograph, is an accurate, safe, and reliable method for BC assessment and was also used to determine fat mass (FM) and fat-free mass (FFM) [31]. Duplicated measures of systolic and diastolic BP were taken after five minutes of seated relaxation using age-appropriate arm cuffs (Pressostabil model, Welch Allyn Inc., Skaneateles Falls, NY, USA). An average was calculated. To adjust for height differences in weight, FM and FFM, body mass index (BMI), fat mass index (FMI), and fat-free mass index (FFMI) were calculated by dividing with the squared height in meters.

A 2 mL venous blood sample was drawn from the children after minimum three hours of fasting. The blood samples were collected in tubes without anticoagulants. Blood glucose and glycosylated haemoglobin (HbA1c) concentrations were determined from the fresh whole blood sample. Glucose was determined using the HemoCue Glucose 201 RT system (HemoCue, Ängelholm, Sweden). HbA1c was determined using a DCCT aligned Quo-Test A1c Analyser (EKF Diagnostics, Cardiff, Wales). Insulin, C-peptide, triglycerides, total cholesterol, and LDL- and HDL-cholesterol were determined from serum. Insulin and C-peptide were determined by module e601 and lipids by module c501 both of the COBAS 6000 analyser (Roche Diagnostics International Ltd., Rotkreuz, Switzerland).

#### 2.2.3. Covariates

Information on child sex and maternal ethnicity, education, and occupation were collected through questionnaires at the birth visit [24]. Gestational age at birth was determined by a trained nurse using the New Ballard Score test [32]. The family’s socioeconomic status was assessed by the International Wealth Index (IWI), which indicates material well-being of households in LMICs and ranges from 0 to 100 [33]. IWI scores were estimated from questionnaires at the birth visit concerning 12 well-being dimensions. Birth weight was measured to the nearest 0.1 g using the built-in scale of an infant PEA POD (PEA POD, Cosmed, Rome, Italy) [24]. Maternal BMI was calculated from maternal height and weight measured to the nearest 0.1 cm and kg [23]. As no pre-pregnancy data were collected, maternal BMI six months postpartum was used to reflect pre-pregnancy BMI. Child age at 5 years was determined from questionnaires at the 5-year visit. Covariates were chosen a priori based on adjustments from previous studies and proposed associations between these and the outcomes of this study. All were considered potential confounders in our linear regression analyses [34,35,36,37,38,39].

### 2.3. Ethical Consideration

Ethical approval was granted from the Jimma University Ethical Review Committee (RPGE/312/2011, 23 December 2008). The study was registered at ISRCTN.com with the number ISRCTN46718296. Measurements were of no harm and of little inconvenience to the mother–child pairs. Written informed consent was obtained from parents or caregivers prior to enrolment.

### 2.4. Statistical Analysis

All descriptive data are presented as mean ± SD for continuous normally distributed variables and median (Q1–Q3) for non-normally distributed continuous variables. Categorical variables are presented as percentages followed by number of participants (*n*).

Associations of BF groups with anthropometry, BC, and cardiometabolic markers were examined in separate models through multiple linear regression analyses. Results from the linear regression analyses are shown as β (95% confidence intervals (CI)). β indicates change in units of the outcome from one of the non-EBF groups to the reference group “Exclusive”. Non-normally distributed variables, triglycerides, insulin, and C-peptide were log-transformed prior to analysis. Results of log-transformed variables are presented as percentwise change due to back-transformation. Estimates with 95% CI are visualised as forest plots. We ran three different models for each outcome, chosen a priori based on covariates stated above. Model 1 adjusted for sex and age at 5 years. Model 2 additionally adjusted for birth weight, gestational age, and maternal BMI. Model 3 additionally adjusted for IWI and maternal ethnicity, education, and occupation. BP outcomes were additionally adjusted for height in all three models due to correlation between height and BP [35]. Data analysis was conducted using the statistical programme R version 3.6.0 (R Foundation for Statistical Computing, Vienna, Austria).

## 3. Results

### 3.1. Study Population

Of the 644 mother–child pairs enrolled, a sub population of 306 children, who attended follow-up at the 6-month and 5-year visits and had data on BF and at least one outcome, were examined in this study (Figure A1). At 6 months, the majority of children were predominantly breastfed (63%), followed by almost exclusively breastfed (21%), exclusively breastfed (10%), and partially breastfed or not at all (6%). The birth weight was on average 3.05 ± 0.40 kg (Table 1). At 5 years, the average BMI, height, FMI, and FFMI were 14.99 ± 1.19 kg/m^2^, 104.3 ± 4.4 cm, 3.86 ± 1.06 kg/m^2^, and 11.15 ± 0.85 kg/m^2^, respectively (Table 2).

### 3.2. The Associations of BF Status at 6 Months with Outcomes at 5 Years

The associations of BF status at 6 months with the outcomes at 5 years from the linear regression analyses are presented in Figure 1 and Figure 2 (see Table A1 and Table A2 for estimates with 95% CI). Results below are presented as β (95% CI).

#### 3.2.1. Anthropometry

At 5 years, compared with the “Exclusive” group, the “Predominantly” group was associated with a shorter stature of 1.7 cm (0.2, 3.3) (model 3). Compared with “Exclusive”, the “Almost exclusive”, “Predominantly” and “Partial or none” groups were associated with a lower BMI of 0.56 kg/m^2^ (0.06, 1.06), 0.55 kg/m^2^ (0.08, 1.01), and 0.89 kg/m^2^ (0.17, 1.61) (model 3). No between-group differences were observed for waist circumference.

#### 3.2.2. Body Composition

Compared with “Exclusive”, children in the “Predominantly” group had a 0.36 kg/m^2^ (0.005, 0.71) lower FFMI, and children in the “Partial or none” group had a 0.81 kg/m^2^ (0.12, 1.50) lower FMI (model 3).

#### 3.2.3. Blood Pressure

Compared with “Exclusive”, the “Almost exclusive”, “Predominantly”, and “Partial or none” groups were associated with a lower systolic BP of 4 mmHg (0.9, 7), 5 mmHg (2, 8), and 4 mmHg (0.06, 9), respectively (model 3). The “Almost exclusive” and “Predominantly” groups were additionally associated with a lower diastolic BP.

#### 3.2.4. Other Cardiometabolic Markers

Compared with the “Exclusive” group, the “Predominantly” group was associated with 53.34% (2.01, 130.49) higher insulin concentrations (model 3). Moreover, compared with the “Exclusive” group, lower total and LDL-cholesterol of 11.7 mg/dL (0.8, 22.5) and 10.2 mg/dL (0.1, 20.4), respectively, were observed among children in the “Almost exclusive” group (model 3). Lower HDL-cholesterol concentrations of 4.7 mg/dL (0.2, 9.3), 4.6 mg/dL (0.5, 8.7), and 8.7 mg/dL (2.3, 15.1), respectively, were also observed among children in the “Almost exclusive”, “Predominantly”, and “Partial or none” groups compared with children in the “Exclusive” group (model 3). No between-group differences were observed for glucose, HbA1c, C-peptide, or triglycerides.

## 4. Discussion

Compared with EBF during the first six months of life, we found non-EBF to be associated with a lower BMI, systolic BP, and HDL-cholesterol at five years of age. Children in the “Predominantly” group were shorter and had a lower FFMI compared with “Exclusive”. The same trend was observed for the “Almost exclusive” group, whereas “Partial or none” compared with “Exclusive” had a lower FMI. Compared with “Exclusive”, lower total and LDL-cholesterol were observed among children in the “Almost exclusive” group, lower diastolic BP was observed among children in the “Almost exclusive” and “Predominantly” groups, and higher insulin was observed among children in the “Predominantly” group.

### 4.1. Anthropometry and Body Composition

At 5 years, the average BMI in the population was close to growth standards [40], and the average height was approximately 1 SD below the height-for-age standard median [41]. The average FMI at 5 years was higher than UK standard scores, and the average FFMI was approximately 2 SD below UK standard scores [42]. The average higher BMI observed among ‘Exclusive’ seems to be largely driven by an accretion in FFM rather than FM as a trend of higher FFMI was observed in this group compared to children in the ‘Almost exclusive’ and ‘Predominantly’ groups. This may suggest that children that are exclusively breastfed for the first six months become taller and have a higher BMI at 5 years. This indicates a positive effect of EBF and that exclusively breastfed children build metabolic capacity (FFM: greater vital organ and muscle mass) rather than metabolic load (FM). A higher accretion of FFM is very important in this specific population, as their average BC is skewed towards a more undesirable composition, with FFMI below standards and FMI above. Previous studies also found children that are breastfed closer to the breastfeeding recommendations to have a higher BMI at 6–11 years [34,43]. One of these studies found that both FMI and FFMI were increased among children that were exclusively breastfed for longer when comparing three durations of exclusive BF: <3 months, 3 to <6 months, and ≥6 months [43]. In accordance with our findings, a higher BMI does not necessarily indicate an unhealthy BC but simply just larger children.

The observed higher BMI in the “Exclusive” group compared specifically with the “Partial or none” group does not, however, seem to be explained by a higher FFMI in the exclusive group but rather by a lower FMI among children in the “Partial or none” group. As this population’s average FM percentage at birth of 7% was low compared with high-income reference data of 10–15% [44,45], a higher FMI in the “Exclusive” group would not necessarily be undesirable. Overall, our results indicate a more desirable BC among children in the “Exclusive” group compared with the non-EBF groups.

### 4.2. Cardiometabolic Markers and Blood Pressure

Consistent with our findings for cholesterol, a previous literature review found a trend of higher concentrations among exclusively breastfed children the first years of life and suggested that higher cholesterol is partly a result of the higher cholesterol concentrations in breast milk compared with substitutes [9]. It has, however, been suggested that high cholesterol exposure in early life can reduce the endogenous synthesis of cholesterol and thereby persist into adulthood, causing lower total and LDL-cholesterol concentrations among those breastfed in infancy [9,14,46,47]. As a result of nutritional programming, children in the “Exclusive” group from the iABC cohort may therefore end up with more beneficial concentrations of these lipids in adulthood. However, the proposed effects of BF on lipid concentrations are based on research from high-income settings and may not apply directly to this low-income setting [14,47]. The average lipid values of the population at 5 years were within normal healthy ranges, except for HDL-cholesterol concentrations, which were below the lower healthy cut-off of 35 mg/dL (Table 2) [48,49]. All non-EBF practices were associated with lower HDL-cholesterol compared with EBF. However, lower total and LDL-cholesterol were only observed among one of the non-EBF practices, and it may seem that the lipid profile of the children in the “Exclusive” group in this study is already more favourable than that of less breastfed children. A UK study also found a lower LDL:HDL ratio among adolescents that were fed human milk as infants compared with formula-fed and suggested a long-term beneficial effect of human milk on the cholesterol metabolism [47]. Also, and in accordance with our findings, a study with Indian children followed until 6 months found that a shift towards a healthy lipid profile began early in life [46]. Moreover, as the average HDL-cholesterol among the children was too low, and the lipid profile thereby trended towards dyslipidaemia, even a small effect size on the individual may still be of importance at the population level [50].

Our findings of lower BP among the “Almost exclusive” and “Predominantly” groups compared with “Exclusive” are contrary to proposed mechanisms that BF has a lowering effects on BP [15]. However, these mechanisms are based on comparison of formula feeding with BF, and we have not examined what the children in our study received instead of breast milk. Moreover, these mechanisms are proposed in a systematic review and meta-analysis that covers a wide range of ages (from children to elderly) and countries (middle- and high-income). A UK study found lower mean BP among adolescents aged 13–16 years who as preterm infants were fed banked human milk compared with preterm formula [51]. This is despite the fact that at follow-up at 8 years, they did not find any BP differences. As also stated by the authors, this suggests that the nutritional programming of BP may not manifest until puberty or beyond, which may explain our findings [51]. Previous research has also shown that higher BP can be caused by a greater height, FMI, and/or FFMI [52]. We adjusted BP for height, but differences in FFMI may contribute as an explanatory component of our findings of higher BP among children in the “Exclusive” group, as we found a lower FFMI among those in the “Predominantly” group and a borderline lower FFMI among those in the “Almost exclusive” group. Importantly, the average BP of the population was within the normal healthy range (Table 2) [35,48,53], and the higher BP among the “Exclusive” group is therefore not abnormal nor harmful.

Our findings of higher insulin among children in the “Predominantly” group compared with “Exclusive” are in agreement with findings from a previous study, which found breastfeeding to be associated with lower insulin concentrations [54]. However, the fact that higher insulin concentrations are only observed among the “Predominantly” and not among the other non-EBF groups could be due to the “Partial or none” group being too small to observe differences and the “Almost exclusive” group being adequately breastfed to obtain the benefits of breastfeeding in this regard. Or, this could simply be a chance finding.

### 4.3. Strengths and Limitations

The strengths of this study include long follow-up with a prospective design reducing the risk of recall bias. There was no recall period as to whether breast milk was the main food given, as the questionnaires covered current breastfeeding behaviour at time of visit. However, a recall of up to six months could occur regarding if water or anything else was given, as questionnaires asked about consumption since birth. This as well as the high-quality BC data is especially unique in regard to what has previously been conducted in low-income settings.

To limit the influence of confounding factors, which are inevitable in observational studies, we adjusted for variables considered potential confounders. However, we were not able to adjust for maternal smoking, as these data were not collected. As rates of smoking are low in Ethiopia, especially among women, this has likely not influenced our findings [55]. Use of BMI six months postpartum as pre-pregnancy BMI may introduce measurement bias. According to previous literature this is approximately when the majority of women aged 25 years return to their pre-pregnancy weight [36]. Exposure misclassification may be present as follow-up might not have been at exactly 6 months of age. However, the deviation from the exact age of the child is most likely not of critical extent. Children that may have been breastfed exclusively for, e.g., 4.5 months, but not at 6 months, will be classified in one of the non-EBF groups despite receiving benefits of breast milk for at least 4.5 months, which may diminish the between group differences. Yet, categorisation into these four BF groups is important as research has shown that even small amounts of supplements can affect nutritional status, morbidity and mortality, and that even provision of water can increase the risk of diarrhoea in these settings [30]. Nondifferential misclassification may have resulted by the short duration of three hours fasting. However, it was not feasible nor justifiable for the small children to have an overnight fast.

Exclusion of half of the enrolled participants due to not attending relevant visits has decreased power. Yet, we observed significant differences between groups indicating a reasonable power for at least some variables [56]. Especially ‘Partial or none’ is a small group, which also is evident from the wide CI’s, and comparisons including this group are therefore less trustworthy compared to findings related to comparisons of the remaining BF groups. The high loss to follow-up may have introduced selection bias. However, a previous study on the iABC cohort found that those who attended and those who did not attend the 5-year visit were largely similar at birth [24].

The iABC cohort is generalisable with other urban Ethiopian populations and other sub-Saharan African urban settings with similar socioeconomic background. The mother-child pairs were restricted to be residing in Jimma Town and recruited from a hospital, where it is more common for urban families to give birth [26]. Furthermore, socioeconomic status of the enrolled families (IWI of 45/100, Table 1) approximated the average from 2012 of Ethiopian urban areas (IWI of 43/100), which is way higher than the Ethiopian average [29] and therefore not generalisable to rural areas. Moreover, only few children in the iABC cohort were breastfed according to the recommendations compared to previous studies from Ethiopia [18,19].

## 5. Conclusions

Our findings suggest that children exclusively breastfed at 6 months of age have a higher BMI at 5 years compared with children only predominantly breastfed. However, the larger size resulted from higher fat-free mass and greater stature but similar fat mass. The larger size at 5 years associated with EBF may therefore be considered beneficial for building long term metabolic capacity without increasing metabolic load. Moreover, exclusively breastfed children had higher HDL-cholesterol and blood pressure and lower insulin compared with predominantly breastfed children. Apart from a more favourable BC and stature, following the breastfeeding recommendations are associated with multiple other benefits for both the child and the mother, such as improved immunity of the child, which is especially important in low-income settings. Future studies are needed to clarify whether the observed differences in our study will persist and whether new benefits of EBF will appear later in life due to programming effects that may not manifest until beyond childhood.

## Figures and Tables

**Figure 1 nutrients-15-04595-f001:**
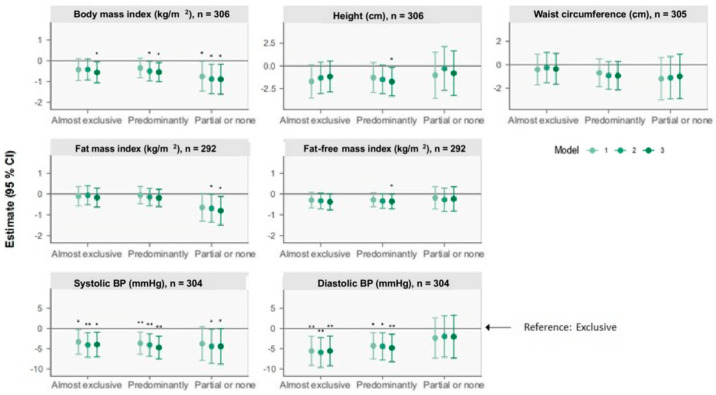
Forest plots of associations between breastfeeding status at 6 months and anthropometry, body composition (BC), and blood pressure (BP) outcomes at 5 years of age retrieved from multiple linear regression analysis. The plots show change (β) in units with 95% CI for outcomes of the nonexclusive breastfeeding groups compared with the reference group, “Exclusive” breastfeeding, illustrated by the horizontal 0-line. Model 1 adjusts for sex and age at the 5-year visit. Model 2 additionally adjusts for birth weight, gestational age, and maternal BMI six months postpartum. Model 3 additionally adjusts for International Wealth Index, ethnicity, education, and occupation at enrolment. Systolic and diastolic blood pressures are additionally adjusted for height at the 5-year visit for all three models. * *p* < 0.05, ** *p* < 0.01.

**Figure 2 nutrients-15-04595-f002:**
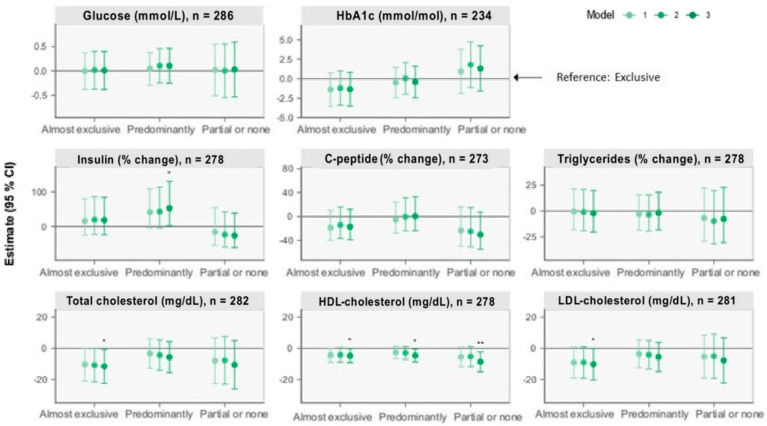
Forest plots of associations between breastfeeding status at 6 months and cardiometabolic markers at 5 years retrieved from multiple linear regression analysis. The plots show change (β) in units with 95% CI for outcomes of the nonexclusive breastfeeding groups compared with the reference group, “Exclusive” breastfeeding, illustrated by the horizontal 0-line. Model 1 adjusts for sex and age at the 5-year visit. Model 2 additionally adjusts for birth weight, gestational age, and maternal BMI six months postpartum. Model 3 additionally adjusts for International Wealth Index, ethnicity, education, and occupation at enrolment. * *p* < 0.05, ** *p* < 0.01.

**Table 1 nutrients-15-04595-t001:** Background characteristics at birth of 306 urban Ethiopian children and their mothers ^1^.

Child Characteristics	
Gestational age (weeks), *n* = 306	39.0 (±1.0)
Weight (kg), *n* = 306	3.05 (±0.40)
Fat mass (%), *n* = 304	7.0 (±4.7)
**Maternal Characteristics**	
Age (y), *n* = 306	24.5 (±4.7)
Height (cm), *n* = 306	157.4 (±6.1)
BMI ^2^ (kg/m^2^), *n* = 300	21.95 (±3.63)
Education level (%)	
No school, *n* = 22	7.2
Some primary school, *n* = 138	45.1
Completed primary school, *n* = 47	15.4
Secondary school, *n* = 59	19.3
Higher education, *n* = 40	13.1
Occupation (%)	
Working, *n* = 92	30.3
Housewife, *n* = 183	60.2
Student, *n* = 25	8.2
Other, *n* = 4	1.3
International Wealth Index ^3^, *n* = 306	45 (±17)
Ethnicity (%)	
Oromo, *n* = 150	49.3
Amhara, *n* = 52	17.1
Other, *n* = 102	33.6

^1^ Data are mean (±SD) for continuous normally distributed variables and percentages for categorical variables. Numbers in categories may not add up, due to missing values. ^2^ Mothers’ BMI from visit 6 (six months postpartum). ^3^ Scale of 0 to 100.

**Table 2 nutrients-15-04595-t002:** Anthropometric, body composition, blood pressure, and cardiometabolic marker variables of 306 urban Ethiopian children at 5 years of age ^1^.

Anthropometry	
BMI (kg/m^2^), *n* = 306	14.99 (±1.19)
Height (cm), *n* = 306	104.3 (±4.4)
Waist circumference (cm), *n* = 305	51.4 (±3.0)
**Body Composition (BOD POD)**	
Fat mass index (kg/m^2^), *n* = 292	3.86 (±1.06)
Fat-free mass index (kg/m^2^), *n* = 292	11.15 (±0.85)
**Blood Pressure, mmHg**	
Systolic, *n* = 304	88 (±7)
Diastolic, *n* = 304	54 (±8)
**Cardiometabolic Markers**	
Glucose (mmol/L), *n* = 286	5.9 (±0.8)
HbA1c (mmol/mol), *n* = 234	37.6 (±4.3)
Insulin (µU/mL), *n* = 278	5.89 (3.27–11.35)
C-peptide (ng/mL), *n* = 273	1.06 (0.67–1.54)
Triglycerides (mg/dL), *n* = 278	85.5 (64.3–114.8)
Total cholesterol (mg/dL), *n* = 282	132.0 (±23.3)
HDL (mg/dL), *n* = 278	30.0 (±9.9)
LDL (mg/dL), *n* = 281	63.7 (±21.7)

^1^ Data are mean (±SD) for continuous normally distributed variables and median (Q_1_–Q_3_) for non-normally distributed variables.

## Data Availability

The iABC Study is part of the Jimma University Clinical and Nutrition (JUCAN) Research Partnership, Jimma, Ethiopia (https://www.ju.edu.et/jucan/ (accessed on 26 October 2023). The iABC study data can be made available upon request directed to the JUCAN Steering Committee, currently headed by Melkamu Berhane (melkamuberhane@yahoo.com). Data cannot be made available in a public repository due to legal and ethical restraints. The informed consent for the iABC study was collected in 2008–2011, and the public sharing of any individual-level data was not part of the consent at that time.

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
