# Peer review of "The Associations of Breastfeeding Status at 6 Months with Anthropometry, Body Composition, and Cardiometabolic Markers at 5 Years in the Ethiopian Infant Anthropometry and Body Composition Birth Cohort"

_nutrients, 2023, doi:10.3390/nu15214595_

Round 1
Reviewer 1 Report
Comments and Suggestions for Authors
1.- In the A1 supplementary text...
Line 437..... says ...Children in the 'Predominantly' group were mainly
breastfed but received complementary food and water as well ("Yes" to question 1 and 3)
Probably ,it should read "Yes" to question 1,2,3.
This should also be corrected in the table.
2.-In table 2 ,the normal values of analytical parameters and BP should be indicated.
3.-The interpretation of the values of triglyceride and, to a lesser extent, cholesterol analyses at 3 hours after fasting should be justified in the comments as plasma triglycerides are elevated by the physiological plasma transport mechanism before 8 hours after intake.
Author Response
Thank you very much for the positive and constructive review.
We have responded to the comments below and revised the manuscript accordingly, and hope it will now be acceptable for publication.
Reviewer 1:
- In the A1 supplementary text …
Line 437…. says …..Children in the ‘Predominantly’ group were mainly breastfed but received complementary food and water as well (“Yes to question 1 and 3”)
Probably, it should read “Yes” to question 1,2,3.
This should also be corrected in the table
Response: Thanks. We agree that this was not clear. Question 2 was not applicable for this group, as children fell into the ‘Predominantly’ group when mothers answered ‘Yes’ to question 3, independent of the answer to question 2.
We have now clarified this in the manuscript in Text A1 by adding the sentence above (Line 439-41). Hence, we kept the table as it is.
- –In table 2 , the normal values of analytical parameters and BP should be indicated.
Response: Thanks for this suggestion. We would prefer not to add the normal values for the following reasons: first, the aim of this paper is to study association among breastfeeding and these outcomes in healthy children only, not in sick individuals. Second, we see that this is not normally done in Nutrients and other journals. Third, it will not look nice, since obviously normal or reference values doesn’t exist for some of the other variables, such as body composition. However, if the editor insist then we will be happy to add it.
- –The interpretation of the values of triglyceride and, to a lesser extent, cholesterol analysis at 3 hours after fasting should be justified in the comments as plasma triglycerides are elevated by the physiological plasma transport mechanisms before 8 hours after intake
Response: The reviewer is right that it would indeed have been optimal with a longer duration of fasting. However, in studies of small children it is usually not possible or justifiable. We have added a sentence about this in the strength and limitation section in the discussion (line 353-356).

Reviewer 2 Report
Comments and Suggestions for Authors
This paper is a large-scale survey study of the importance of breastfeeding in low-income contry.
The paper is recognized as an adequate study, as it simultaneously analyzes birth information, education, and other relevant factors.
The detailed discussion of the results obtained and their consistency and differences with previous reports is highly commendable.
We do not require any major revisions. Please consider the following
Please clarify the inclusion and exclusion criteria (do all participants live in low-income countries?).
Why were preterm infants or low birth weight infants excluded in this study?
Were family environment (father's education), standard of living (including household income), and regional differences investigated?
Regarding the method of blood pressure measurement; please specify whether resting blood pressure was measured, blood pressure only at blood collection, or average blood pressure
Author Response
Thank you very much for the positive and constructive review.
We have responded to the comments below and revised the manuscript accordingly, and hope it will now be acceptable for publication.
Reviewer 2:
- This paper is a large-scale survey study of the importance of breastfeeding in low-income contry. The paper is recognized as an adequate study. As it simultaneously analyzes birth information, education and other relevant factors. The detailed discussion of the results obtained and their consistency and differences with previous reports is highly commendable. We do not require any major revisions. Please consider the following
Response: Thanks for the positive comments.
- Please clarify the inclusion and exclusion criteria (do all participants live in low-income countries?)
Response: As written (Line 84-87), the iABC cohort “enrolled mothers residing in Jimma town, Ethiopia, and their apparently healthy newborns, born at term with a birth weight ≥1500g. The newborns were examined within 48 h after delivery at Jimma University Hospital and were followed up continuously until 5 years of age”.
We have now added “(≥37 weeks of pregnancy)” (Line 86) to make that clear.
We have furthermore added a sentence about the generalisability of the study population in the discussion (Line 366-368).
- Why were preterm infants and low birth weight infants excluded in this study?
Response: We deliberately excluded preterm children and very low-birth weight children because the aim of the study was to study normal and healthy children. Furthermore, it would not have been possible to bring such small and vulnerable children to the facilities where the Peapod was.
- Were family environment (father’s education), standard of living (including household income), and regional differences investigated?
Response: Unfortunately, we did not have data on father’s education. However, Table 1 gives an indication of background characteristics, including maternal education. There were no regional differences, since all were residing in Jimma town. But as mentioned in our discussion (Line 366-369), standard of living (indicated by average IWI) is higher in Jimma Town, compared to rural areas of Ethiopia.
- Regarding the method of blood pressure measurement: please specify whether resting blood pressure was measured, blood pressure only at blood collection, or average blood pressure.
Response: Blood pressure at five years was measured after 5 minutes of seated resting, the same day as the other measurements - blood collection was the last element in the data collection process. The blood pressure was measured in duplicates and an average was calculated (Line 114-116).
